# Discovery and Development of Inhibitors of the Plasmodial FNT-Type Lactate Transporter as Novel Antimalarials

**DOI:** 10.3390/ph14111191

**Published:** 2021-11-20

**Authors:** Cornelius Nerlich, Nathan H. Epalle, Philip Seick, Eric Beitz

**Affiliations:** Department of Pharmaceutical and Medicinal Chemistry, Pharmaceutical Institute, Christian-Albrechts-University of Kiel, Gutenbergstr. 76, 24118 Kiel, Germany; cnerlich@pharmazie.uni-kiel.de (C.N.); nepalle@pharmazie.uni-kiel.de (N.H.E.); pseick@pharmazie.uni-kiel.de (P.S.)

**Keywords:** formate–nitrite transporter, lactate, *Plasmodium*, malaria, antimalarials

## Abstract

*Plasmodium* spp. malaria parasites in the blood stage draw energy from anaerobic glycolysis when multiplying in erythrocytes. They tap the ample glucose supply of the infected host using the erythrocyte glucose transporter 1, GLUT1, and a hexose transporter, HT, of the parasite’s plasma membrane. Per glucose molecule, two lactate anions and two protons are generated as waste that need to be released rapidly from the parasite to prevent blockage of the energy metabolism and acidification of the cytoplasm. Recently, the missing *Plasmodium* lactate/H^+^ cotransporter was identified as a member of the exclusively microbial formate–nitrite transporter family, FNT. Screening of an antimalarial compound selection with unknown targets led to the discovery of specific and potent FNT-inhibitors, i.e., pentafluoro-3-hydroxy-pent-2-en-1-ones. Here, we summarize the discovery and further development of this novel class of antimalarials, their modes of binding and action, circumvention of a putative resistance mutation of the FNT target protein, and suitability for in vivo studies using animal malaria models.

## 1. Introduction

Malaria remains one of the most prevalent human parasitic diseases, being responsible for 229 million recorded infections and 409,000 deaths in 2019, mainly of children under the age of five [1]. The most severe and widespread form of cerebral malaria is caused by *Plasmodium falciparum*. *P. vivax* prevails outside of Africa, and *P. knowlesi*, *P. malariae*, and *P. ovale* appear to be more locally restricted. Despite encouraging results toward vaccination, disease control, and eradication, resistance of *Plasmodium* spp. parasites to antimalarial drugs is on the rise, stressing the need to widen our therapeutic arsenal with new drugs [2,3].

Malaria parasites are protozoa that multiply within erythrocytes in which they undergo complex developmental transformations over 2–3 days. The malaria-typical recurrent fever symptoms are elicited by synchronized rupturing of infected erythrocytes and evasion of the parasites in the merozoites state [4]. After new infection of erythrocytes, the parasites transform into the ring state, followed by the metabolically most active trophozoite state before they form schizonts that split into evading merozoites. Intra-erythrocytic malaria parasites rely on anaerobic glycolysis consuming glucose from the infected host to meet their energetic requirements (Figure 1) [5,6,7]. d-glucose is taken up via the erythrocyte’s glucose transporter, GLUT1 [8,9], and the parasite’s hexose transporter, HT [10,11]. The protein structures of the human [12,13] and recently of the plasmodial glucose transporters [14] have been resolved at high resolution. Such transporters are themselves attractive targets for the design of antimalarials [15,16,17]. In fact, the HT structure and a human homolog, GLUT3, were elucidated in the presence of an inhibitor, C3361, and the structures were used successfully to optimize the affinity of the small molecule [13,14]. Such compounds exhibit antimalarial activity in parasite cultures by blocking glycolysis via HT-inhibition [13,14,15,17] or by inducing apoptosis via redox stress when acting on the erythrocyte GLUT [16].

From each mole of d-glucose, the parasite generates two moles of ATP, as well as two moles of l-lactate and protons as metabolic end-products [18]. In order to avoid cytosolic acidification, plasmodia swiftly export lactate and protons [19,20,21]. Despite biochemical knowledge of the plasmodial energy metabolism as well as of lactate and proton release, the responsible transport protein remained elusive for decades. Recently, we and others identified a single *P. falciparum* gene that encodes a formate–nitrite transporter-type protein (PfFNT) to act as the missing lactate/proton co-transporter of the parasites [18,22]. The strictly microbial FNT proteins share no sequence similarity with the human monocarboxylate transporters (MCT) [23]. With the discovery of PfFNT, the transport components of the malaria parasite’s energy flux were complete (Figure 1).

Sequence comparison of the FNT proteins from the five *Plasmodium* species that infect humans (*P. falciparum*, *P. vivax*, *P. malariae*, *P. ovale*, and *P. knowlesi*) showed > 84% overall similarity with the internal substrate transport path, with the isoforms being virtually identical [24]. The vital functionality and lack of similarity with the human MCTs appeared to be beneficial for the design of PfFNT-specific antimalarials. FNT isoforms are present in other human-pathogenic parasites, such as *Toxoplasma gondii* [25] and *Entamoeba histolytica* [26].

PfFNT is a homopentameric membrane protein (Figure 2, left) [27,28,29]. Each protomer acts as an individual bidirectional transport unit (Figure 2, center) [30]. The N- and C-termini are located at the cytoplasmic side, and the protomer fold consists of six transmembrane helices around a symmetrical narrow transport path [28,29,30,31,32]. Two lipophilic constrictions isolate a highly conserved central histidine residue from wider vestibules at both entrance sites (Figure 2, right). The FNT protein surface is characterized by a positive electrostatic potential that attracts and funnels lactate anions into the vestibules. The increasingly lipophilic environment inside the vestibules facilitates protonation of the entering lactate anions to form neutral lactic acid, allowing the substrate to pass the lipophilic constriction sites [26,31,32,33].

Here, we describe the development process of small-molecule inhibitors that partially resemble the transport substrate and make use of the selectivity and transport mechanisms yet potently block *Plasmodium* spp. FNTs. The compounds kill malaria parasites at nanomolar concentrations in vitro. The inhibitor design further includes a structural feature aiming at the prevention of resistance formation, i.e., a PfFNT G107S point mutation, which we observed after forced selection by a non-lethal dosing regime [34].

## 2. Structures of Small Molecule PfFNT Inhibitors

### 2.1. Initial Weak Inhibitors Hinted at the Therapeutic Potential of Targeting Plasmodial Lactate Transport

The antimalarial activity of known lactate transport inhibitors, such as cinnamic acid derivatives, the bioflavonoid phloretin, the loop-diuretic furosemide, and niflumic acid, was described prior to the discovery of PfFNT [19,20,21]. However, millimolar concentrations of cinnamic acid derivatives were required to kill cultured *P. falciparum* parasites [19]. After the identification and cloning of PfFNT, we determined respective IC_50_ values of about 1 mM using PfFNT-expressing *Saccharomyces cerevisiae* yeast exposed to a 1 mM inward lactate gradient [18]. Apparently, the tested inhibitors and the substrate competed with similar affinity for PfFNT binding. Lactate uptake of isolated *P. falciparum* parasites in the metabolically most active trophozoites state was decreased by 70% after treatment with 100 µM niflumic acid or 5-nitro-2-(3-phenylpropylamino)benzoic acid (NPPB) [22]. The inhibition of lactate/H^+^ transport gave rise to acidification of the *Plasmodium* cytoplasm triggering a cascade of detrimental effects, i.e., cell-swelling and blockage of proton gradient-driven transmembrane transport events among others, which were responsible for the ultimate death of the parasite [35].

The identified compounds have a weak acid moiety in common (pK_a_ range of 3–4.5), which was found to be a requirement for inhibition by replacing the carboxy group with the respective amide [18]. The compounds most likely act as substrate analogues that approach and bind the PfFNT protein in the same way as the lactate anion (pK_a_ of lactic acid: 3.8) yet are too large to pass the transduction pathway (see Figure 3 for a structural comparison of lactic acid **[1]** and α-fluorocinnamic acid **[2]**).

Knowing that the blockade of plasmodial lactate transport represents a valid antimalarial approach in principle, the discovery of the specific PfFNT target protein prompted library screenings for effective drug-like inhibitors.

### 2.2. The MMV Malaria Box Contains Two Potent PfFNT Inhibitors

The Malaria Box provided by the Medicines for Malaria Venture (MMV) is a collection of 400 drug-like compounds derived from phenotypic screenings of *P. falciparum* parasite cultures that address unknown targets with an EC_50_ < 4 µM [36]. Screening of these compounds yielded two hits that directly inhibited PfFNT—MMV007839 **[3]** and MMV000972 **[4]**—differing only in the presence or absence of an aromatic methoxy substituent (Figure 4) [37].

Two independent screening strategies led to the identification of the PfFNT inhibitors [35,37]. We expressed PfFNT in a yeast strain that lacks endogenous lactate transporters and assayed the uptake of ^14^C radiolabeled l-lactate in the presence of the individual Malaria Box compounds at 10 µM [37]. An alternative screening setup used fluorescent dye-loaded *P. falciparum* parasites to detect a decrease in the cytosolic pH upon treatment with the compounds followed by a radiolabel transport assay with PfFNT-expressing *Xenopus laevis* oocytes [35]. The assays yielded nanomolar IC_50_ values for direct PfFNT target inhibition. The EC_50_ values obtained with cultured 3D7 strain *P. falciparum* parasites were at 0.14 µM for MMV007839; yet MMV000972 appeared less potent on the living parasites by an order of magnitude (1.7 µM) [35,37]. The difference in the in vitro efficacy was attributed to the uptake of the compounds into the parasite.

MMV007839 and MMV000972 contain hemiketal moieties that are prone to hydrolysation. Indeed, using correlation NMR, we found an equilibrium of the hemiketal form with the respective vinylogous acid (Figure 4). The vinylogous acid tautomers share some resemblance with the weakly PfFNT-inhibiting cinnamic acid derivatives (Figure 3) and represent the actual binding forms of the identified hit compounds. We showed this by removal of the phenol hydroxyl moiety, which prevented hemiketal formation, yet was perfectly tolerated without a decrease in activity in the yeast lactate transport assay. The neutral hemiketal form may thus facilitate permeation across the consecutive lipid membranes of an infected erythrocyte (Figure 1), whereas the deprotonated, negatively charged vinylogous acid anion binds PfFNT and blocks lactate transport [37].

Studies on the structure–activity relationships based on MMV007839 as a lead established the first generation of specific PfFNT inhibitors (Figure 4) [37]. The presence of a vinylogous acid moiety turned out to be essential for PfFNT inhibition, as was a halogenated alkyl chain. Changes in the length of the fluoroalkyl chain showed that pentafluoroethyl was optimal, while elongation to heptafluoropropyl yielded more active compounds than shortening to trifluoromethyl. Therefore, further structure optimization focused on substituents of the aromatic ring and replacement of the benzene itself. Here, the *para*-position provided the highest degree of flexibility for modifications. A moderate replacement of methoxy by ethoxy in MMV007839 increased the in vitro efficacy to an EC_50_ of 50 nM [37]. Together, these analyses led to the pentafluoro-3-hydroxy-pent-2-en-1-one-scaffold and the pharmacophore **[5]**, which already exhibited a surprisingly high potency with an IC_50_ of 1.9 µM in the yeast PfFNT transport assay considering its small size (Figure 4) [37].

### 2.3. Forced Resistance Selection Revealed the Binding Site of PfFNT Inhibitors

Treatment of *P. falciparum* in vitro cultures with sub-lethal doses of MMV007839 selected for resistant parasites with a shifted EC_50_ by two orders of magnitude from 0.14 µM to 35 µM [35,37]. Subsequent sequencing of the PfFNT-encoding gene from the resistant parasite strain displayed a single point mutation resulting in an amino acid exchange of Gly107 by serine. The decrease in activity of MMV007839 was attributed to a direct effect by the PfFNT G107S mutation using the mutated target in yeast transport assays. The transport capability of PfFNT G107S for the lactate substrate, however, was only slightly affected [37].

With the known location of Gly107 in the PfFNT transport path based on structure models at that time (Figure 2), we concluded that the binding site of MMV007839 resides in the cytoplasmic vestibule and constriction region. Further taking the structure–activity relationships and PfFNT transport mechanism into account, we proposed a binding mode for MMV007839 (Figure 5).

According to the model, major interactions occur between the hydrophobic fluoroalkyl chain of the inhibitor and the lipophilic cytoplasmic constriction of PfFNT and via the vinylogous acid moiety and Thr106 [37]. The polarity of the binding site is clearly different from that of the glucose transporters, which is composed of mainly hydrophilic amino acid residues to match the polar characteristics of a sugar molecule [13,14]. Further, a slim, linear shape is required in the pharmacophore part of the PfFNT inhibitor to fit the narrow transport path of PfFNT, while more space is available in the area of the aromatic ring and in particular at the *para*-position due to the orientation towards the wider transporter entrance site. It is conceivable that the pharmacophore scaffold mimics two daisy-chained lactate molecules and manifests two phases of the underlying PfFNT transport mechanism. Accordingly, the fluoroalkyl chain would represent the neutral protonated lactic acid passing the lipophilic constriction, and the vinylogous acid moiety would reflect the entering negatively charged lactate anion. The aromatic ring shields the interaction sites from the aqueous bulk, increasing the affinity. In the PfFNT G107S resistance mutant, the serine sidechain with its hydroxyl moiety protrudes into the transport path, rendering it narrower. We suspected that the phenol hydroxyl group of MMV007839 collides with the serine (Figure 5) and figured that an adaptation of the inhibitor structure to circumvent the clash might be possible [37].

### 2.4. Circumvention of the PfFNT G107S Resistance Mutation by Introduction of Scaffold Nitrogen Atoms

Simple removal of the phenol hydroxyl from MMV007839 increased the activity against PfFNT G107S by one order of magnitude (BH296 **[6]**, IC_50_ 2.3 µM; Figure 6) [37]. However, the efficacy of BH296 was still one order of magnitude lower for the G107S mutant than for wildtype PfFNT (IC_50_ 0.14 µM). Further, the lack of a phenol hydroxyl group prevents cyclization to the neutral hemiketal (Figure 4), which may affect the in vitro efficacy due to impeded transmembrane passage of the more polar vinylogous acid. Indeed, the EC_50_ obtained with 3D7 *P. falciparum* parasites was in the micromolar range (3.6 µM). A breakthrough in addressing the PfFNT G107S mutation was reached by introducing nitrogen atoms into the aromatic ring to provide hydrogen bond acceptor sites for interaction with the serine hydroxyl moiety (Figure 6) [34]. One of the resulting compounds (BH297 **[7]**; Figure 6) achieved similar nanomolar inhibition of PfFNT G107S (0.26 µM) and the wildtype protein (0.11 µM). Nevertheless, the in vitro EC_50_ of BH297 with 3D7 *P. falciparum* parasites was still in the micromolar range (3.88 µM). Removal of the BH297 methoxy group resulted in BH267.meta **[8]** and finally achieved submicromolar potency for inhibition of PfFNT wildtype (0.11 µM), PfFNT G107S (IC_50_ = 0.63 µM), as well as an in vitro EC_50_ of 0.29 µM; Figure 6) [34]. The positioning of the nitrogen in the heteroaromatic ring was crucial. Only the meta and ortho positions enabled efficient hydrogen bond formation with a serine replacement at position 107 of PfFNT. Surprisingly, the in vitro efficacies of BH267.meta **[8]** and BH267.ortho were quite different (Figure 6), possibly indicating separate uptake routes into the parasite, which requires further investigation. Determination of true binding kinetics and affinities using fluorescence cross-correlation spectroscopy with solubilized PfFNT showed good correlation with the yeast data [38]. Another promising outcome of the development of BH267.meta was that, contrary to MMV007839, resistance formation in cultures of 3D7 *P. falciparum* parasites appeared to be suppressed.

Furthermore, BH267.meta showed very low cytotoxicity towards human cells [24] and minimal off-target potency on the human lactate transporter, MCT1 [24,37]. Cell viability as determined by ATP quantity and resazurin reduction and cell proliferation measured by nuclei count was tested using human kidney (HEK293) and liver (HepG2) cell lines. In all assays, BH267.meta showed negligible toxicity at the highest tested concentration of 100 μM, i.e., three orders of magnitude higher than its IC_50_ value. The effect of BH267.meta on MCT1 was low at an IC_50_ of around 500 μM.

### 2.5. The PfFNT Cryo-Electron Microscopy Structure Confirms the Proposed Binding Mode

Very recently, the protein structure of PfFNT was revealed as a complex with the MMV007839 inhibitor (Figure 7) [28,29]. The data confirm that the vinylogous acid represents the binding tautomer. The binding site in the cytoplasmic vestibule and constriction region, as well as the interacting amino acids, were found to be correctly predicted in our models. Eventually, the proposed clash of the hydroxyl moieties from the PfFNT G107S resistance mutation with MMV007839, and the resolution of the collision by removal of the phenyl hydroxyl and formation of a hydrogen bond instead with the BH267.meta compound were found to be highly plausible as deduced from modeling (Figure 7) [29].

## 3. Conclusions

Lactate transport inhibitors with a pentafluoro-3-hydroxy-pent-2-en-1-one-scaffold are a valid novel class of antimalarials with a new mechanism of action. The target, PfFNT, is druggable and has no structural counterpart in humans. The compounds inhibit FNT isoforms from all five human-pathogenic *Plasmodium* species; they exhibit minimal off-target effects on the human lactate transporter MCT1 [24,37], and very low cytotoxicity in mammalian cell lines [24]. Introduction of scaffold nitrogen atoms circumvents the G107S resistant mutation [34]. Overall, BH267.meta is a promising candidate to progress to in vivo studies using animal malaria models.

## Figures and Tables

**Figure 1 pharmaceuticals-14-01191-f001:**
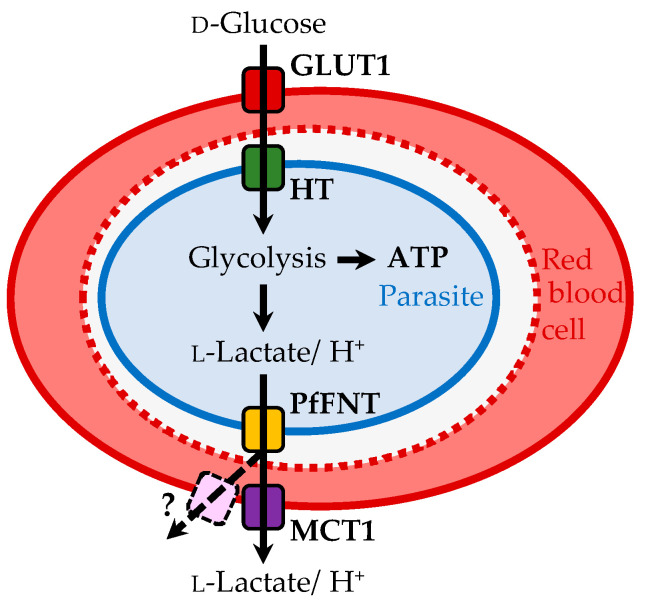
Energy flux in a *P. falciparum*-infected erythrocyte. Glucose is transported via the erythrocyte glucose transporter 1, GLUT1, and the hexose transporter, HT, into the parasite’s cytoplasm. Glycolysis generates ATP from glucose, forming the metabolic end products l-lactate and protons that are exported via PfFNT into the erythrocyte cytosol, and eventually by the human monocarboxylate transporter, MCT1, or alternative red cell export pathways.

**Figure 2 pharmaceuticals-14-01191-f002:**
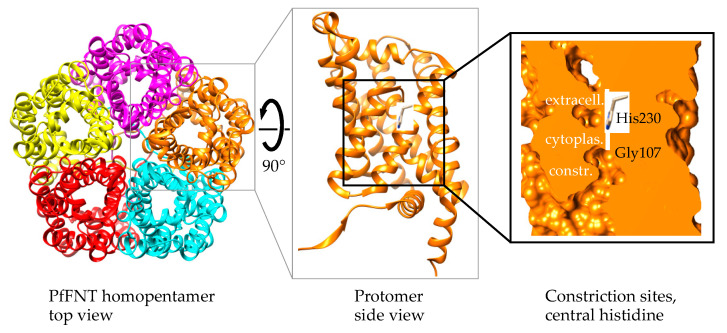
PfFNT protein structure. Shown are the homopentamer (PDB# 7e26) as seen from the extracellular side (left), a protomer in side view, and a space-fill display of the internal transport pathway with two constriction sites that sandwich a highly conserved, central histidine residue.

**Figure 3 pharmaceuticals-14-01191-f003:**
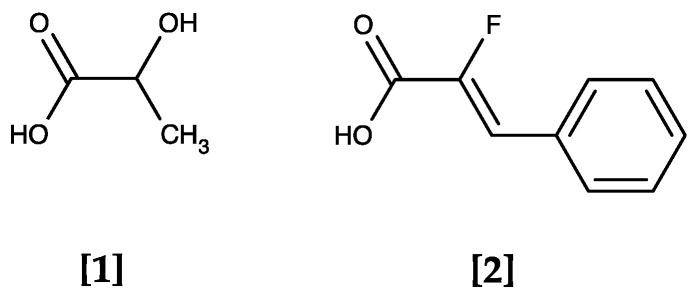
The PfFNT transport substrate lactic acid **[1]** and α-fluorocinnamic acid **[2]**, a weak PfFNT inhibitor. Both molecules are weak monocarboxylic acids that largely deprotonate at physiological pH forming the acid anions lactate and α-fluorocinnamate.

**Figure 4 pharmaceuticals-14-01191-f004:**
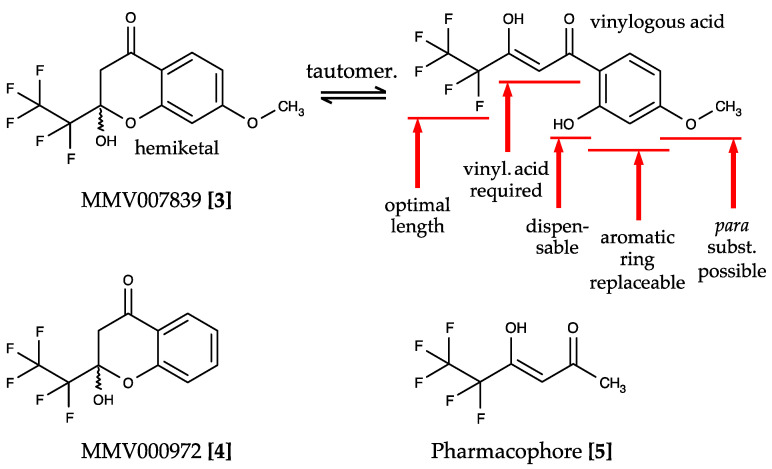
Screening hits of the Malaria Box and structure–activity relationships for PfFNT inhibition. MMV007839 **[3]** and MMV000972 **[4]** are neutral hemiketals that undergo tautomerization to form weak vinylogous acids in solution. Elucidation of the essential moieties and possible modification sites of the compounds identified the pharmacophore **[5]**.

**Figure 5 pharmaceuticals-14-01191-f005:**
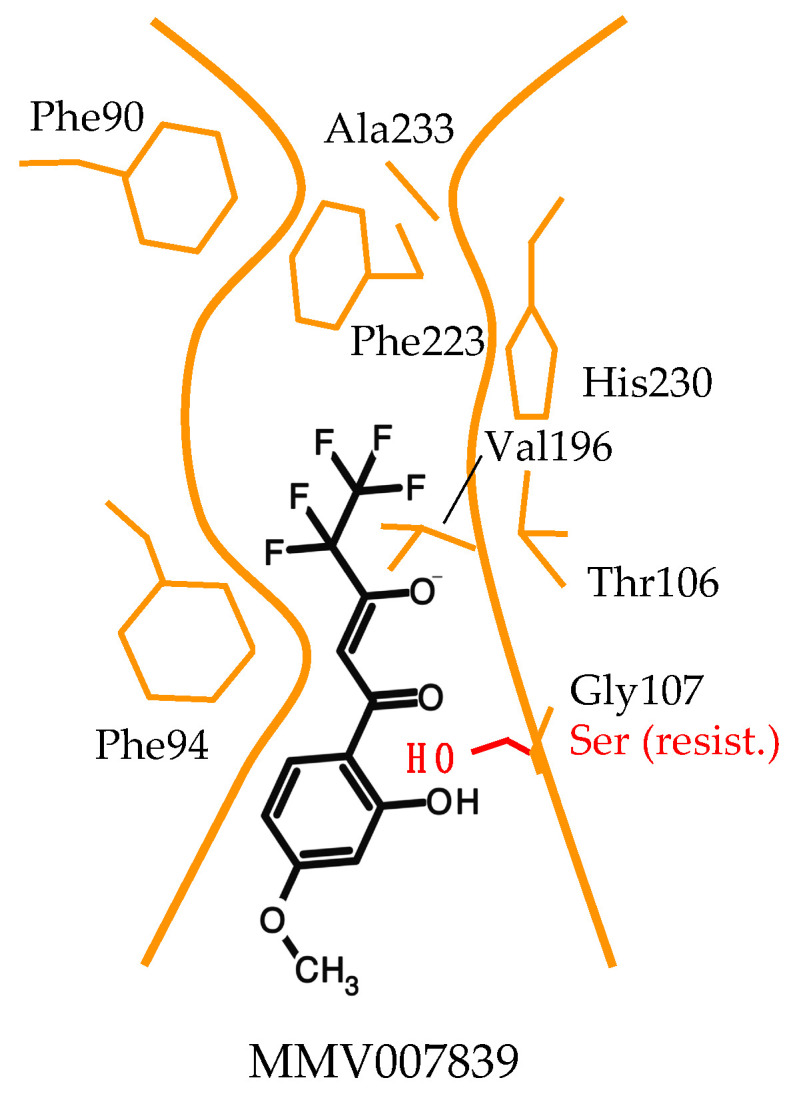
Model of the binding mode of MMV007839 to PfFNT. The inhibitor binds in the cytoplasmic vestibule and constriction region. The observed G107S resistance mutation produces a clash with the phenol hydroxyl of MMV007839.

**Figure 6 pharmaceuticals-14-01191-f006:**
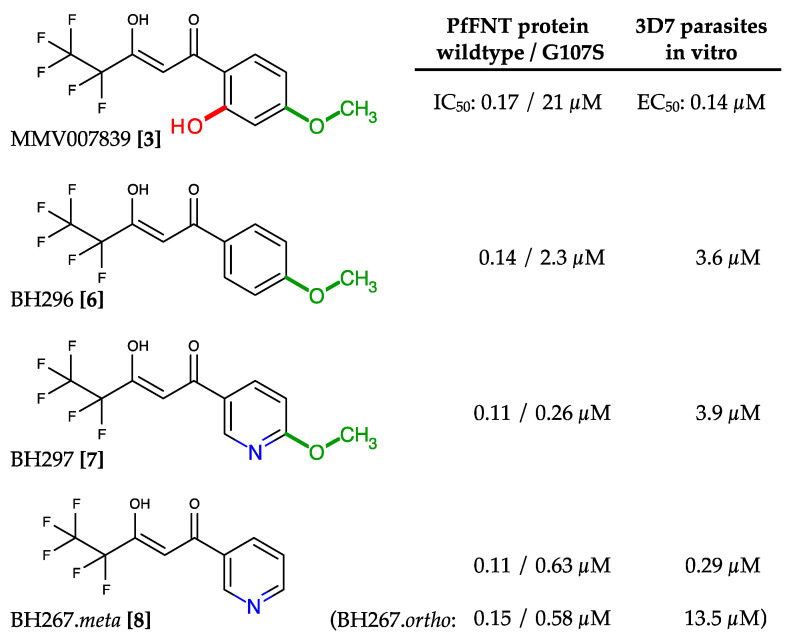
Development of PfFNT inhibitors with activity against the G107S resistance mutation. The clashing phenol hydroxyl and the methoxy moieties were successively removed, and nitrogen atoms were introduced into the aromatic to act as hydrogen bond acceptor sites. Eventually, with BH267.meta, a compound was generated exhibiting nanomolar efficacy on the PfFNT wildtype and G107S mutant protein, as well as in 3D7 *P. falciparum* parasite in vitro cultures.

**Figure 7 pharmaceuticals-14-01191-f007:**
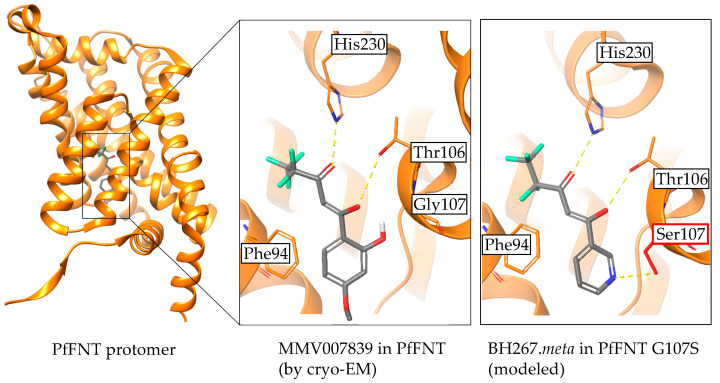
Confirmation of the binding mode of PfFNT inhibitors by cryo-electron microscopy. Visualization of MMV007839 bound to PfFNT was done using PDB# 7e27 with the Maestro software (Release 2021-3; Schrödinger, New York, USA). The display of BH267.meta in PfFNT G107S was generated with the Protein Preparation Wizard and subsequent molecular docking using the standard, ligand centered Induced Fit Docking protocol.

## Data Availability

Data sharing not applicable.

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
