# Peer review of "Discovery and Development of Inhibitors of the Plasmodial FNT-Type Lactate Transporter as Novel Antimalarials"

_pharmaceuticals, 2021, doi:10.3390/ph14111191_

Round 1

Reviewer 1 Report

The Review by Cornelius Nerlich and colleagues recapitulate the work focused on modulators of Plasmodial FNT-Type Lactate Transporter as potential antimalarial agents. The topic is covered in an adequate manner, the relevant literaturature has been cited and overall is a well written review. I have no modification to suggest

Author Response

Point: "The Review by Cornelius Nerlich and colleagues recapitulate the work focused on modulators of Plasmodial FNT-Type Lactate Transporter as potential antimalarial agents. The topic is covered in an adequate manner, the relevant literaturature has been cited and overall is a well written review. I have no modification to suggest."

Response: We appreciate the comment.

Reviewer 2 Report

The review is well focused and contains up-to-date information on FNT-Type Lactate Transporter.

The only thing that seems to me that in order to be a review, the authors could expand the information related to the use of glucose by parasites and thus connect their way of obtaining the energy necessary for their development.

The authors speak of the interaction of the inhibitory molecules, but it could be previously explained what type of amino acids are in the channel that form this receptor GLUT1.

In this way, more importance is given to the existence of this type of inhibitor.

Author Response

Point 1:  “The only thing that seems to me that in order to be a review, the authors could expand the information related to the use of glucose by parasites and thus connect their way of obtaining the energy necessary for their development.”

Response: We added a paragraph (lines 37-46) introducing the human GLUT and parasite HT transporters for glucose, their structure, small-molecule inhibitors, and antimalarial action on glycolysis and probably apoptosis. The text is substantiated by six new references (lines 327-342).

Point 2: “The authors speak of the interaction of the inhibitory molecules, but it could be previously explained what type of amino acids are in the channel that form this receptor GLUT1. In this way, more importance is given to the existence of this type of inhibitor.”

Response: 

We now address the internal amino acid composition and provide a comparison between GLUT and PfFNT transporters regarding the substrate/inhibitor binding site (lines 200-202).

Reviewer 3 Report

The review article “ Discovery and Development of Inhibitors of the Plasmodial FNT-Type Lactate Transporter as Novel Antimalarials ” by Nerlich et. al. is suitable for publication after addressing the following minor comments:

Since cytotoxicity is also important along with potency, authors should provide some published cytotoxicity data of the PfFNT inhibitors in mammalian cell lines, and also authors should comment on in vivo data of the PfFNT inhibitors if published in the literature.

Author Response

Point 1: “Since cytotoxicity is also important along with potency, authors should provide some published cytotoxicity data of the PfFNT inhibitors in mammalian cell lines, and also authors should comment on in vivo data of the PfFNT inhibitors if published in the literature.”

Response: We added text with more detailed information on the low cytotoxicity and off-target effects of the BH267.meta inhibitor (lines 260-266). There is no published in vivo efficacy data available.